# Comparison of Regular and Speed Sintering on Low-Temperature Degradation and Fatigue Resistance of Translucent Zirconia Crowns for Implants: An In Vitro Study

**DOI:** 10.3390/jfb13040281

**Published:** 2022-12-08

**Authors:** Suchada Kongkiatkamon, Chaimongkon Peampring

**Affiliations:** Department of Prosthetic Dentistry, Faculty of Dentistry, Prince of Songkla University, Hat Yai 90110, Songkhla, Thailand

**Keywords:** all-ceramic, zirconia, dental materials, firing, sintering, speed sintering, prosthetic dentistry, biaxial flexural strength, Weibull

## Abstract

Background: Although there are a few studies which compare fast and slow sintering in normal zirconia crowns, it is essential to compare the cracks and load-bearing capacity in zirconia screw-retained implant crowns between regular and speed sintering protocols. This research aimed to compare the surface structure, cracks, and load-bearing capacity in zirconia screw-retained implant crowns between regular sintering (RS) and speed sintering (SS) protocol with and without cyclic loading (fatigue). Methods: A total of 60 screw-retained crowns were fabricated from zirconia (Katana STML Block) by the CAD/CAM system. Then, 30 crowns were subjected to the RS protocol and 30 crowns were subjected to the SS protocol. Cyclic loading was done in half zirconia crowns (15 crowns in each group) using a chewing simulator CS-4.8/CS-4.4 at room temperature. The loading force was applied on the middle of the crowns by a metal stylus underwater at room temperature with a chewing simulator at an axial 50 N load for 240,000 cycles and lateral movement at 2 mm. Scanning electron microscopy was done to study the surface of the crowns and the cracks in the crowns of the regular and speed sintering protocols, with and without fatigue. Results: For the speed sintering group, the surface looks more uniform, and the crack lines are present at a short distance compared to regular sintering. The sintering protocol with a larger Weibull module and durability increases the reliability. It showed that the Speed group showed the maximum fracture load, followed by the regular, speed fatigue, and regular fatigue groups. The fracture load in various groups showed significant differences. Conclusions: It was found that the speed group showed the maximum fracture load followed by the regular, speed fatigue, and regular fatigue. The crack lines ran from occlusal to bottoms (gingiva) and the arrest lines were perpendicular to the crack propagations.

## 1. Introduction

Zirconia ceramics are the choice of materials for esthetic dental restorations and prosthetic crowns and they are also commonly used for the fabrication of implant crowns [1]. Recently, the translucency of zirconia materials has improved, resulting in more esthetic restorations in prosthetic dentistry [2,3]. In addition, zirconia has high strength and high wear resistance, can resist masticatory forces, and is widely used for the stress-bearing posterior teeth [4,5,6,7]. Zirconia restorations also presents good biocompatibility [8,9]. Currently, the use of digital technologies, computer-aided design, and computer-aided manufacturing (CAD/CAM) have allowed the simplified production of precise and durable implant components resulting in high clinical success rates and outcomes [10,11]. The precise fit of the crown is affected by the design of the implant [12,13,14].

Translucent zirconia can be obtained by increasing translucency and decreasing opacity. Optimal translucency in zirconia can be obtained by increasing grain size, optimizing grain boundary region, and increasing yttria content to produce partially stabilized zirconia with increased amounts of nonbirefringent cubic phase [15,16]. In addition, the opacity in zirconia can be decreased by sintering additives (i.e., alumina), reducing oxygen vacancies, defects, and pores, and controlling the sintering environment (pressure and temperature) [15,17]. It was found that as the translucency in zirconia increased, the mechanical properties (especially the flexural strength) were reduced [18].

Zirconia is subjected to various processes such as milling, grinding, and polishing during the manufacturing process. From these processes, surface and volume flaws are formed which differ according to the location, size, and orientation, resulting in variable strength [19,20,21,22]. These process-related microstructures and flaws can affect the fracture strengths of the restoration. Furthermore, the zirconia can undergo low-temperature degradation (LTD) and can result in damage to implant restoration [23].

Zirconia stabilized with 5 mol% yttria content (5Y-TZP) and possesses excellent mechanical properties and biocompatibility [24,25]. Although there are few studies which compare fast and slow sintering in normal zirconia crowns, it is essential to compare the cracks and load-bearing capacity in zirconia screw-retained implant crowns between regular and speed sintering protocols, and there are no studies focused on this comparison. Hence, this research compared the surface structure, cracks, and load-bearing capacity in zirconia screw-retained implant crowns fabricated from third generation zirconia under regular and speed sintering protocols, with and without cyclic loading (fatigue).

## 2. Materials and Methods

### 2.1. Construction of Screw-Retained Zirconia Crowns

The study overview is shown in Figure 1. A total of 56 sample sizes is calculated from the G*Power software with a power of 90%. A total of 60 screw-retained zirconia crowns were fabricated from the zirconia material (Katana STML Block as shown in Table 1) by the CAD/CAM system as explained by Nogueira et al. [26]. At first, dental implants (Astra Tech EV, diameter 4.2 mm) were placed in the dentoform at the mandibular first molar region. The implants were placed at the center of the tooth and platform 3 mm below the gingival line. All crowns were of the same size (4 mm thick measured from the Ti base abutment outer surfaces and 1.74 mm measured from the top of the Ti base abutment).

Scan post and scan body were utilized to transfer the information and position to the CAD software by using inLAb CAD Software 22.0 (Dentsply Sirona, Germany) (Figure 2). Ti-base GH 1 mm was selected. Sixty Crowns were milled by using MCXL and then sintered by using speed sintering (SS) protocol (*n* = 30) and regular sintering (RS) protocol (*n* = 30).

The zirconia screw-retained crowns were fabricated from zirconia block, then scanned by Cerec Omicam (Dentsply Sirona, Bensheim, Germany), designed by inLAb CAD Software 22.0 (Dentsply Sirona, Bensheim, Germany), and finally milled with an MCXL (Dentsply Sirona, Bensheim, Germany) milling unit (Figure 2). Then the crowns were sintered following the respective manufacturer’s recommendations. For the RS group, the total thermal cycle, sintering time, and dwell temperature were 6.8 h, 2 h, and 1550 °C, respectively, by using inFire HTC Furnace (Dentsply Sirona, Bensheim, Germany). For the SS group, the total thermal cycle, sintering time, and dwell temperature were calculated based on each specimen, which was 30 min, 16 min, and 1560 °C, respectively, by using SpeedFire oven (Dentsply Sirona, Bensheim, Germany) [28]. In each group, half of the crowns were again subjected to fatigue using LTD by putting the autoclave at 122 °C under −2 bar pressure for 8 h, and half of the crowns were again subjected to no fatigue.

### 2.2. Cementation of Screw-Retained Zirconia Crowns

All crowns were cemented to Ti-base. Ti base and Zirconia crowns were sandblasted (50-micron 1 bar) for 5 s. Single Bond Universal (3MESPE, Saint Paul, MN, USA) was utilized to apply both crowns and Ti-base surfaces. Rely X Ultimate (3MESPE, Saint Paul, MN, USA) was utilized to cement following the manufacturer’s instructions. Screw-retained crowns were torqued at 25 Ncm to implant analog. Teflon tape and resin composite Z350 (3MESPE, Saint Paul, MN, USA) were used to close access screw holes of 2.6 mm diameter.

### 2.3. Cyclic Loading of Screw-Retained Zirconia Crowns

The zirconia crowns (15 crowns in each group) were subjected to cyclic loading using the chewing simulator CS-4.8/CS-4.4 (CS 4.8, SD Mechatronic GmbH, Feldkirchen-Westerham, Germany) at room temperature. The technical specifications included traverse paths of the Z-axis approx. 120 mm, and X-axis approx. 50 mm, speed ranged between 1–60 mm/s in both axes, dimensions including thermocycling unit was 190 × 200 × 80 cm (length × height × depth), and weight load of maximum 10 kg per sample chamber. The loading force was applied on the middle of the crowns by a metal stylus underwater at room temperature with a chewing simulator at an axial 50 N load for 240,000 cycles (frequency of 1 Hz) and lateral movement at 2 mm (Figure 3A) [29].

Then, the surface, cracks, and load-bearing capacity of the zirconia crowns were studied under regular and speed sintering protocols with and without fatigue.

### 2.4. Load to Failure

All zirconia crowns (fatigued and non-fatigued) were subjected to a single load-to-failure using the universal testing machine (Lloyd instruments, Model LRX-Plus, AMETEK Lloyd Instrument Ltd., Hampshire, UK) according to ISO 6872 [30]. An indenter consisting of a steel ball (6 mm diameter) was aligned at the same contact point as the dynamic loading with an increasing speed of 1 mm/min. Cracks on zirconia crowns and crown fractures (chipping cohesive fractures and/or bulk catastrophic fractures) were analyzed (Figure 3B).

### 2.5. Surface Characterization and Fractographic Analysis

The surface structures of the zirconia specimens were characterized using scanning electron microscopy (SEM) (FEI Quanta 400, FEI, Czech) of zirconia specimens (3 samples for each group). The specimens were submitted to sputter-coating with a gold-palladium alloy. The images were observed with an accelerating voltage of 20 kV in various magnifications. Selected crowns of each group were studied for the fractographic analysis with SEM (EVO10, Carl Zeiss, Oberkochen, Germany).

### 2.6. Weibull’s Analysis

For the zirconia crowns, a probabilistic approach was done to analyze the strength and lifetime. The fracture strength of zirconia is done by the Weibull formula as shown in Equation (1) [31]:(1)$$P_{f}=1-\exp[-\left(\sigma_{c}-\sigma_{0}{)}^{m}\right]$$
where P*_f_* = fracture probability, σ*_0_* = characteristic strength, σ*_c_* = fast-fracture strength without subcritical crack growth, and m = Weibull modulus which is the slope of the regression line in the lnσ*_c_* − ln ln [1/1 – P*_f_*)] diagram [32]. σ*_0_* relates to the stress point where 63.2% of the crowns would fracture. The fast-fracture strength of each crown was calculated by the formula shown in Equation (2) [20]:(2)$$\mathrm{In}\;\mathrm{In}\frac{1}{1-P_{f}}=m\;\mathrm{In}\;\sigma_{c}-m\;\mathrm{In}\sigma_{0}.$$

### 2.7. Statistical Analysis

Descriptive statistics were calculated, and the results were compared among the sintering groups. The statistical analysis was done using SPSS 20.0 (SPSS Inc., IBM Corporation, Chicago, IL, USA) to see the significant differences (*p* value = 0.05). One-way ANOVA with multiple comparisons using the Scheffe analysis was done to compare the results among various groups.

## 3. Results

### 3.1. Fractographic Analysis

The surface of various zirconia crowns of various sintering groups is shown in Figure 4. Figure 4 shows that, for all samples, the crack lines run from the top (occlusal) to the bottom (gingiva). Indeed, the arrest lines are perpendicular to the crack propagations. For the SS group, the material looks more uniform, and the crack lines are present at shorter distances compared to RS; this concurs with the result that SS provides higher strength than RS. Once the material was fatigued, more crack lines were presented. The direction of crack lines in the fatigue group is presented in different directions compared to the non-fatigue group. In this study, all zirconia crowns presented with bulk catastrophic material as shown in Figure 5.

### 3.2. Load to Failure

The descriptive statistics of the fracture load are shown in Table 2. It shows that the speed group showed the maximum fracture load, followed by the regular, speed fatigue, and regular fatigue groups. The multiple comparisons show that the fracture load was a significant difference among the various groups; regular vs. regular fatigue, regular vs. speed, regular vs. speed fatigue, regular fatigue vs. speed, regular fatigue vs. speed fatigue, and speed vs. speed fatigue (Table 3).

Figure 6 shows the Weibull distribution for the zirconia crowns. The 63.3% value was also marked, for which the strength according to the analysis is σ*_0_* = 3244 MPa for regular, σ_0_ = 2190 MPa for regular fatigue, σ*_0_* = 3765 MPa for speed, and σ*_0_* = 2457 MPa for speed fatigue (Table 4). It shows that only the regular group shows an R^2^ of less than 65%. The sintering protocol with a larger Weibull module and durability increases the reliability.

## 4. Discussion

The zirconia restorations are subjected to the occlusal loads but are below the flexural strength of zirconia in the mouth. However, the pre-existing surface flaws may lead to the propagation of those cracks and eventually cause the fracture of the zirconia restoration [15,33]. We analyzed the zirconia’s survival using the Weibull approach. A low Weibull modulus shows more variability in the strength (σ_c_), whereas a high Weibull modulus signifies higher reliability of zirconia material. Weibull’s theory is based on the weakest link theory whereby failure is the result of the weakest line break in the chain [32].

The crack propagation mechanism in zirconia materials differs according to the type of material composition and sintering protocols. For the two material types, it was found that an intergranular fracture is seen in 3Y-TZP and a combination of intergranular and intragranular crack propagation is seen in 5Y-TZP [20]. When a micro-crack is created during the manufacturing process, the failure will start.

Our results were similar to the results obtained by some previous studies [34,35,36]. Mayinger et al. [34] studied the impact of high-speed sintering and artificial aging on the fracture load of three-unit zirconia fixed dental prostheses and found promising results for high-speed sintering, as it led to comparable fracture load and similar, or even superior, Weibull modulus results compared to the control group. The 4Y-TZP material presented fracture load results similar to the tried-and-tested 3Y-TZP. Artificial aging did not influence zirconia’s resistance to fracture for either 3Y-TZP or 4Y-TZP. Similarly, Jerman et al. [36] investigated the influence of high-speed and conventional sintering on the flexural strength of three zirconia materials, initially and after artificial aging, and found that high-speed sintered HT+ showed higher initial flexural strength than the control group (*p* < 0.001). ZI (*p* < 0.001–0.004) and Zolid (*p* < 0.001–0.007) showed higher flexural strength after thermomechanical aging. In addition, the Weibull modulus of the three thermomechanically aged materials was negatively influenced by high-speed sintering; hence, they also mentioned that high-speed sintering is a valid alternative to conventional sintering protocols.

Arcila et al. [37] studied the microstructure of 3 yttria partially stabilized (3Y-PSZ) disc-shaped zirconia and compared the fracture resistance, hardness, and fatigue flexural strength of 3Y-TZP, 4Y-PSZ, and 5Y-PSZ. They demonstrated that all zirconia materials show similar compositions but vary the yttria content (5Y-PSZ > 4Y-PSZ > 3Y-TZP). They concluded that despite the microstructural differences, 3Y-TZP and 4Y-PSZ showed similar fatigue behavior whereas 5Y-PSZ had the least fatigue behavior.

Ordoñez Balladares et al. [38] compared the fracture strength of monolithic Zr dioxide after sintering in two different furnaces: CEREC SpeedFire (fast sintering) and InFire HTC Speed (slow sintering) produced from CAD/CAM. They found that the CEREC SpeedFire presented 1222.8 N fracture strength, whereas InFire HTC Speed showed 1068.5 N fracture strength, but no significant differences among the groups. The different furnaces did not affect the strengths of the zirconia and there is time saving when using rapid sintering. However, speed-sintered restorations may have limited reliability. According to a previous article, super-speed sintering is considered an option [28]; however, in the esthetic area where translucency is required, super-speed sintering might not suitable.

The thickness reduction of zirconia and fatigue affects the failure load of monolithic zirconia crowns. Prott et al. [39] mentioned that less thickness of the crown leads to less strength, even if the failure loads surpass the chewing forces. Fatigue significantly lessened the failure load of 0.5 mm 3Y-TZP crowns.

Regarding the limitations of this research, we should note the limited sample size. For the regular group, the R^2^ might be higher if we cut out the samples or add more samples. This research is preliminary. Further research can be carried out to analyze of the chemical chain or strengthening phenomena in zirconia.

## 5. Conclusions

Surface flaws are the failure origin, and the crack lines ran from the occlusal to the bottom. Handling of the zirconia plays an important role in the longevity of the zirconia restorations. It was found that the speed sintering group showed the maximum fracture load followed by the regular, speed fatigue, and regular fatigue groups. These values provide reference fracture load values for the implant crown and fixed partial denture and are used to assess the durability of zirconia in dentistry.

## Figures and Tables

**Figure 1 jfb-13-00281-f001:**
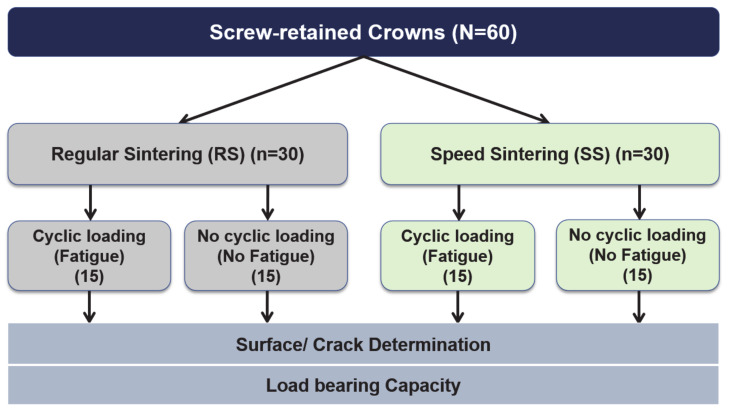
Overview of the study.

**Figure 2 jfb-13-00281-f002:**
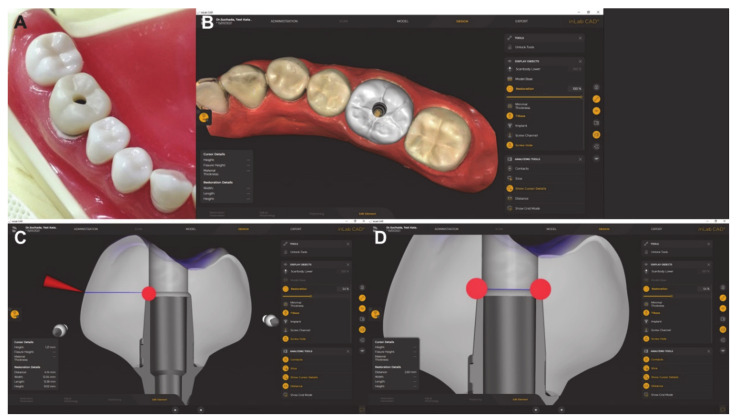
Screw-retained zirconia crowns designed from Omnicam software and completed screw-retained zirconia crown. (**A**) = study model, (**B**) = occlusal view in the software, (**C**) = sectional view of the implant crown, and (**D**) = screw hole measurement.

**Figure 3 jfb-13-00281-f003:**
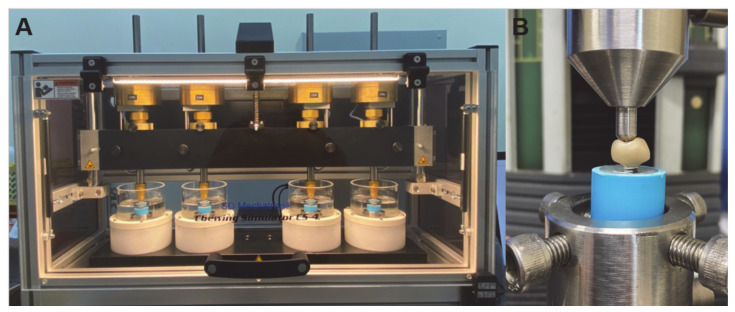
Chewing stimulator and flexural strength testing. (**A**) = Chewing stimulator and machine, (**B**): universal testing machine and set up of biaxial test method for the zirconia disc.

**Figure 4 jfb-13-00281-f004:**
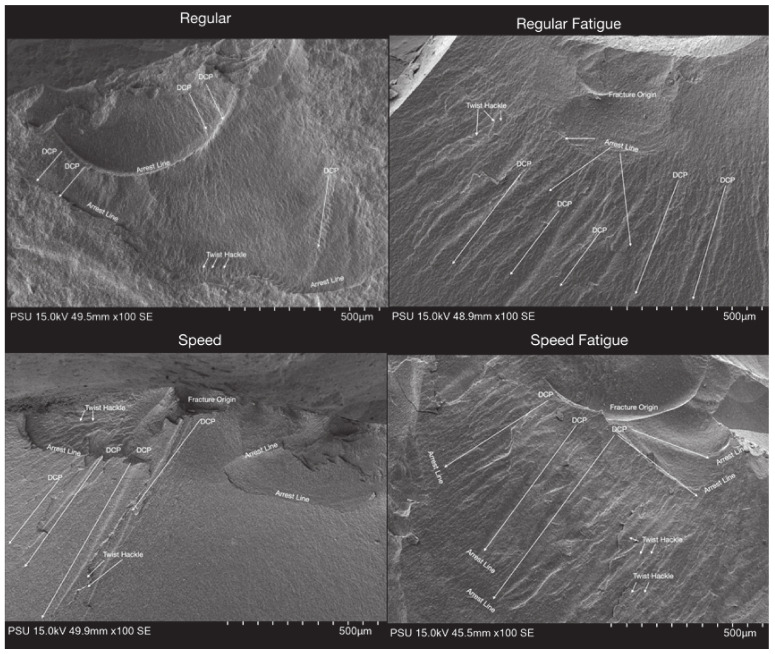
Surface structure in SEM of the zirconia crowns of various sintering groups.

**Figure 5 jfb-13-00281-f005:**
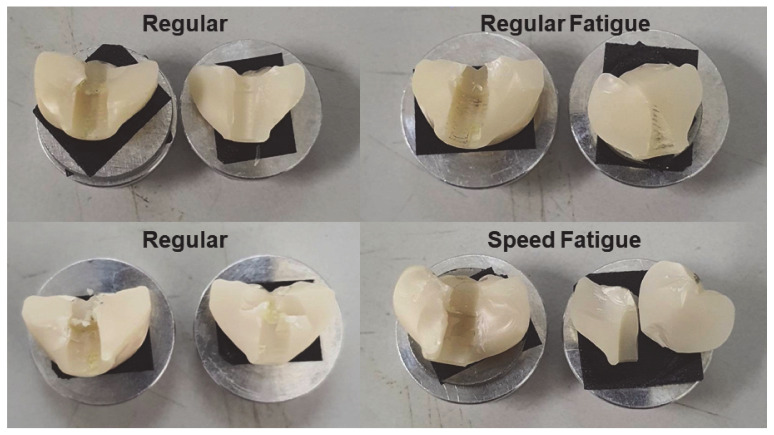
Fracture of the zirconia crowns of various sintering groups.

**Figure 6 jfb-13-00281-f006:**
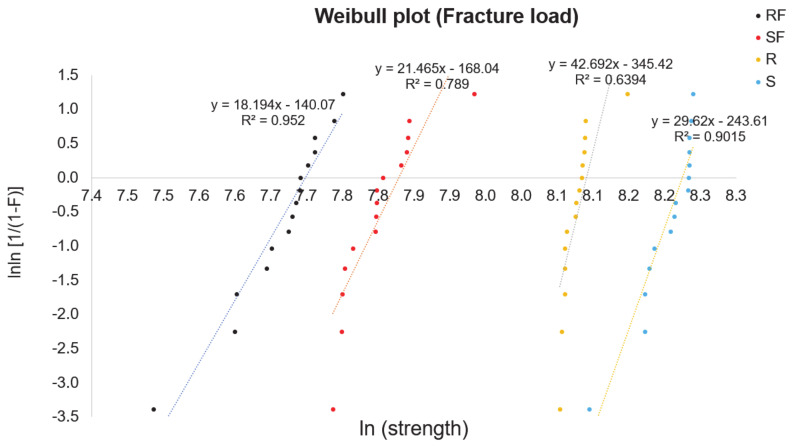
Weibull plot for the strength of the zirconia crowns of various sintering groups. R = regular, S = speed sintering, RF = regular fatigue, SF = speed fatigue.

**Table 1 jfb-13-00281-t001:** Details of the zirconia used in this study [25,27].

Composition	
ZrO_2_ + HfO_2_	88–93 wt.%
Y_2_O_3_	7–10 wt.%
Other oxides	0–2 wt.%
**Properties**	
Flexural strength	748 MPa
Translucency	38%
Coefficient of thermal expansion	9.8 ± 0.2 × 10^−6^ K^−1^

**Table 2 jfb-13-00281-t002:** Descriptive statistics of fracture load.

Sintering	N	Mean (N)	Std. Deviation	Std. Error	Minimum	Maximum
Regular	15	3223.60	76.11	19.65	3146.00	3456.00
Regular fatigue	15	2143.06	135.93	35.09	1783.00	2324.00
Speed	15	3664.46	140.90	36.38	3276.00	3787.00
Speed fatigue	15	2450.06	128.26	33.12	2290.00	2790.00

**Table 3 jfb-13-00281-t003:** Multiple comparisons of the fracture load among the groups.

(I) Groups	(J) Groups	Mean Difference (I-J)	*p* Value
Regular	Regular fatigue	1080.53	<0.001 *
	Speed	−440.86	<0.001 *
	Speed fatigue	773.53	<0.001 *
Regular fatigue	Regular	−1080.53	<0.001 *
	Speed	−1521.40 *	<0.001 *
	Speed fatigue	−307.00	<0.001 *
Speed	Regular	440.86	<0.001 *
	Regular fatigue	1521.40	<0.001 *
	Speed fatigue	1214.40	<0.001 *
Speed fatigue	Regular	−773.53	<0.001 *
	Regular fatigue	307.00	<0.001 *
	Speed	−1214.40	<0.001 *

* The mean difference is significant at the 0.05 level.

**Table 4 jfb-13-00281-t004:** Results of Weibull distribution analysis.

Sintering Protocol	N	Characteristic Strength σ_o_ [MPa]	Coefficient of Determination R2	Weibull Modulus m
Regular (R)	15	3244	0.64	42.69
Regular fatigue (RF)	15	2190	0.95	18.19
Speed (S)	15	3765	0.90	29.62
Speed fatigue (SF)	15	2457	0.79	21.46

## Data Availability

The data presented in this study are available on request from the corresponding authors.

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
