# Peer review of "Comparison of Regular and Speed Sintering on Low-Temperature Degradation and Fatigue Resistance of Translucent Zirconia Crowns for Implants: An In Vitro Study"

_jfb, 2022, doi:10.3390/jfb13040281_

Round 1

Reviewer 1 Report

The article is interesting. The presented results appear to be well discussed and have enough quality to continue to publication.

Author Response

Response to Reviewer 1 Comments

The article is interesting. The presented results appear to be well discussed and have enough quality to continue to publication.

Thank you for your positive comments.

Reviewer 2 Report

This research aimed to compare the surface, cracks, and load-bearing capacity in zirconia screw-retained implant crowns in regular and speed sintering protocol with and without cyclic loading (fatigue). No novel information can be found in this study. The design is weak, and the results cannot support the conclusion. Besides, there are too many defects in the article, some are list as follows:

1. The introduction is not coherent and sufficient enough, so it is suggested to revise the whole content.

2. Figure 2 has no detailed caption. For example, what do A, B, and C stand for respectively?

3. In the method, the author mentioned that the surface structure (2.3) and fractographic analysis (2.6) of the zirconia specimens were characterized by SEM. However, in the results, the description of the surface structure results of zirconia samples and the pictures of fractographic analysis results were misplaced, so the pictures of the surface structure results and the description of the fractographic analysis results of zirconia samples were lacking. It is suggested that the author carefully check the manuscript before submitting it.

4. The author's title includes the research on the influence of low temperature degradation on 5Y-TZP zirconia crowns, but there is no relevant research in the author's article. The author seems to replace the LTD test with the cyclic loading test, which is only carried out at 37℃, but I don’t think it is right.

5. It is suggested to modify the first paragraph of the discussion, which is a repetitive description of the results and meaningless.

6. It is suggested that the author revise the whole content of the discussion. Most of the content in the discussion is descriptions of others’ reports rather than related discussions on the author’s research results, which have no obvious direct supporting relationship with the author's research object.

Author Response

Response to Reviewer 2 Comments

This research aimed to compare the surface, cracks, and load-bearing capacity in zirconia screw-retained implant crowns in regular and speed sintering protocol with and without cyclic loading (fatigue). No novel information can be found in this study. The design is weak, and the results cannot support the conclusion. Besides, there are too many defects in the article, some are list as follows:

  1. The introduction is not coherent and sufficient enough, so it is suggested to revise the whole content.

Response: Thank you for your positive comments. Corrections in the Manuscript for Reviewer 2 are highlighted in Yellow color.

In the Introduction, more content are added.

  1. Figure 2 has no detailed caption. For example, what do A, B, and C stand for respectively?

Response: Details are added in the Figure 2 Caption.

  1. In the method, the author mentioned that the surface structure (2.3) and fractographic analysis (2.6) of the zirconia specimens were characterized by SEM. However, in the results, the description of the surface structure results of zirconia samples and the pictures of fractographic analysis results were misplaced, so the pictures of the surface structure results and the description of the fractographic analysis results of zirconia samples were lacking. It is suggested that the author carefully check the manuscript before submitting it.

Response: The surface characterization and d fractographic analysis are corrected.

  1. The author's title includes the research on the influence of low temperature degradation on 5Y-TZP zirconia crowns, but there is no relevant research in the author's article. The author seems to replace the LTD test with the cyclic loading test, which is only carried out at 37℃, but I don’t think it is right.

Response: We did not use cyclic loading instead of low thermal degradation. In this article, we were trying to simulate the scenario that all specimens that received cyclic loading and then compare the group with and without LTD would give the difference. Hence, described how we simulate the LTD by putting the autoclave at 122 celsius under - 2 bar pressure for 8 hours.

So, we did both.

  1. It is suggested to modify the first paragraph of the discussion, which is a repetitive description of the results and meaningless.

Response: The first paragraph of the discussion is edited.

  1. It is suggested that the author revise the whole content of the discussion. Most of the content in the discussion is descriptions of others’ reports rather than related discussions on the authors research results, which have no obvious direct supporting relationship with the author's research object.

Response: More discussion is edited and revised.

Reviewer 3 Report

The authors showed higher mechanical strength of 5Y-TZP samples sintered using a speed sintering oven than regular sintering. Although the conclusion is based on the results, the mechanism to enhance mechanical properties is unclear. I recommended some improvements that should be made before publication.

1.      The company does not show the Katana STML as 5Y-TZP. Reference #12 doesn't seem to show the character of Katana STML. Please indicate the characterization of Katana STML or show the proper reference.

2.      Please show the details of the sintering condition; heating rate, holding temperature, holding time, and so on.

3.      The legends in Figure 4 are unclear because the font size is too small.

4.      Please show the unit of Mean in Table 1.

5.      It’s unclear why the speed sintering enhanced the mechanical property of 5Y-TZP. The authors should carefully discuss that using proper references.

Author Response

Response to Reviewer 3 Comments

The authors showed higher mechanical strength of 5Y-TZP samples sintered using a speed sintering oven than regular sintering. Although the conclusion is based on the results, the mechanism to enhance mechanical properties is unclear. I recommended some improvements that should be made before publication.

Thank you for your positive comments. Corrections in the Manuscript for Reviewer 1 are highlighted in Green color.

  1. The company does not show the Katana STML as 5Y-TZP. Reference #12 doesn't seem to show the character of Katana STML. Please indicate the characterization of Katana STML or show the proper reference.

Response: Thank you for your positive response. We modified the manuscript per your recommendation.

The composition of Katana STML had 7-10% by weight of yttria according to company’s information.

Reference. https://www.thejpd.org/article/S0022-3913(21)00356-5/pdf

They stated that Katana STML had 5% mole of yttria.

However, we modified our manuscript and made sure the information would give correct information.

  1. Please show the details of the sintering condition; heating rate, holding temperature, holding time, and so on.

Response: Details are added in the Method.

  1. The legends in Figure 4 are unclear because the font size is too small.

Response: The legends are described more in Figure 4.

  1. Please show the unit of Mean in Table 1.

Response: Unit (Newton) is added in Table 1.

  1. It’s unclear why the speed sintering enhanced the mechanical property of 5Y-TZP. The authors should carefully discuss that using proper references.

Response: More discussion is added.

Reviewer 4 Report

Dear Editor,

Regarding the submitted manuscript “  Comparison of Regular and Speed Sintering on Low Temperature Degradation and Fatigue Resistance of 5Y-TZP Zirconia Crowns for Implants.”  this review will report a detailed appreciation.

The presented study is intended to be an in vitro study to evaluate the fatigue resistance of implant supported crowns.

The article is well written, and the proposed objectives were attained. 

It is this referee belief that some changes are needed before considering the manuscript for publication:

1-    Some misspellings were found through the manuscript and should be corrected

2-    Title should be revised, shortened and the “in vitro study” added. 

3-    Abstract: in the results section authors should state results and not only the statistical significance. 

4-    Introduction: at the end of the introduction the hypothesis should be stated.

5-    When looking into the material and methods section some corrections and clarifications are needed:

a.     Sample size determination – authors need to mention the power calculation to address the differences between groups, and calculation of the sample size based on that assumption.

b.     Aging/Cyclic loading – authors need to state and use references to the proposed method of aging (number of cycles, without different water temperatures exposure, etc).

6-    Results

a.     The authors need to mention how many samples were submitted to SEM analysis.

b.      The authors need to present the results regarding cracks on zirconia crowns and crown fractures (chipping cohesive fractures and/or bulk catastrophic fractures.

7-    Discussion:

a.     In the discussion authors should be careful in some of the results addressed since extrapolations should not be performed between in vitro results to in vivo.

b.     The last paragraph should be removed since it is an extrapolation that cannot exist based in this study

Although with a small sample size, the study design is well performed for the evaluated variable , but some changes are needed for publication.

Author Response

Response to Reviewer 4 Comments

Dear Editor, Regarding the submitted manuscript “ Comparison of Regular and Speed Sintering on Low Temperature Degradation and Fatigue Resistance of 5Y-TZP Zirconia Crowns for Implants.”  this review will report a detailed appreciation.

The presented study is intended to be an in vitro study to evaluate the fatigue resistance of implant supported crowns.

The article is well written, and the proposed objectives were attained. 

It is this referee belief that some changes are needed before considering the manuscript for publication:

Thank you for your positive comments. Corrections in the Manuscript for Reviewer 1 are highlighted in Turquoise color.

1-    Some misspellings were found through the manuscript and should be corrected

Response: They are corrected.

2-    Title should be revised, shortened and the “in vitro study” added. 

Response: The title is corrected.

3-    Abstract: in the results section authors should state results and not only the statistical significance. 

Response: The results are edited and improved.

4-    Introduction: at the end of the introduction the hypothesis should be stated.

Response: The title is corrected.

5-    When looking into the material and methods section some corrections and clarifications are needed:

  1. Sample size determination – authors need to mention the power calculation to address the differences between groups, and calculation of the sample size based on that assumption.

Response: A total of 56 sample size is calculated from the G*Power software. We added it in the manuscript. But studied on the 60 samples. (Section 2.1)

  1. Aging/Cyclic loading – authors need to state and use references to the proposed method of aging (number of cycles, without different water temperatures exposure, etc).

Response: Details are added in the Method. (Section 2.1)

6-    Results

  1. The authors need to mention how many samples were submitted to SEM analysis.

Response: Three samples per group. We added it in the manuscript. (Section 2.5)

  1. The authors need to present the results regarding cracks on zirconia crowns and crown fractures (chipping cohesive fractures and/or bulk catastrophic fractures.

Response: Type of failure details with Figures are added in the Manuscript. (Figure 5)

7-    Discussion:

  1. In the discussion authors should be careful in some of the results addressed since extrapolations should not be performed between in vitro results to in vivo.

Response: The Discussion is edited and improved.

  1. The last paragraph should be removed since it is an extrapolation that cannot exist based in this study

Response: The last paragraph is edited, and limitation is added based on comments from other reviewers.

Although with a small sample size, the study design is well performed for the evaluated variable , but some changes are needed for publication.

Response: Thank you. The manuscript is improved.

Reviewer 5 Report

Dear Authors

Thank you for the opportunity to familiarize yourself with the methodology and results of your research. The work seems to me very synthetic, using a simplified model. 

1.     You should state the limitations of this methodology in your paper.

2.     I would recommend having your work read by a native speaker to improve readability.

3.    Indeed, the abstract should be improved as it is very extensive and seems to repeat in detail the materials and methods section of the main section of the manuscript.

4.    When giving the justification for the conducted research, you concluded that similar research has been carried out for prosthetic crowns but not implant crowns (line 53, 54). Please elaborate on why you expect different results and why your research is important to conduct.

5.     You wrote that you introduced Astra implants (line 63). You did not specify whether the implants were placed in the patient or in the model. The work shows that it is probably a model, but you do not describe its type and the resulting limitations. You have not provided the parameters of the reconstructed missing tooth, which may significantly affect the shape of the crown (dimensions and shape of the occlusal surface) with all the resulting consequences.

6.     The dimensions of the crowns were not specified, especially their thickness and whether they were the same.

7.     The holes in crowns for screwing the implant were not specified.

8.     You have not entered a legend for pictures A, B, C in Figure 2

9.     Is it correct to describe Sirona Company in different way in the line 74 and 75?

10.  How long did the sandblasting described in line 82 take? 

11.  On line 141 you wrote that the cracks were located in a smaller area, this is too general a statement and not supported by data

12.  The conclusions should be rebuilt because they are too numerous and almost identical with the introduction and the results.

I hope that clarifying these issues will increase the scientific value of your publication.

Best regards

Author Response

Response to Reviewer 5 Comments

Dear Authors

Thank you for the opportunity to familiarize yourself with the methodology and results of your research. The work seems to me very synthetic, using a simplified model.

Thank you for your positive comments. Corrections in the Manuscript for Reviewer 1 are highlighted in Gray color.

  1. You should state the limitations of this methodology in your paper.

Response: Limitations are added at the end of the Discussion.

  1. I would recommend having your work read by a native speaker to improve readability.

Response: English correction is done. If needed we will use the MDPI English correction service.

  1. Indeed, the abstract should be improved as it is very extensive and seems to repeat in detail the materials and methods section of the main section of the manuscript.

Response: The abstract is edited and improved.

  1. When giving the justification for the conducted research, you concluded that similar research has been carried out for prosthetic crowns but not implant crowns (line 53, 54). Please elaborate on why you expect different results and why your research is important to conduct.

Response: The justification of this research is mentioned at the end of the introduction.

  1. You wrote that you introduced Astra implants (line 63). You did not specify whether the implants were placed in the patient or in the model. The work shows that it is probably a model, but you do not describe its type and the resulting limitations. You have not provided the parameters of the reconstructed missing tooth, which may significantly affect the shape of the crown (dimensions and shape of the occlusal surface) with all the resulting consequences.

Response: Details are added (section 2.1)

  1. The dimensions of the crowns were not specified, especially their thickness and whether they were the same.

Response: Details are added (section 2.1)

  1. The holes in crowns for screwing the implant were not specified.

Response: Details are added (screw hole diameter 2.6 mm) (section 2.1)

  1. You have not entered a legend for pictures A, B, C in Figure 2

Response: The title is corrected.

  1. Is it correct to describe Sirona Company in different way in the line 74 and 75?

Response: It is corrected in RS and SS sintering methods (section 2.1).

  1. How long did the sandblasting described in line 82 take? 

Response: Sandblasting was done for 5 sec (section 2.2).

  1. On line 141 you wrote that the cracks were located in a smaller area, this is too general a statement and not supported by data

Response: This is edited (section 3.1).

  1. The conclusions should be rebuilt because they are too numerous and almost identical with the introduction and the results. I hope that clarifying these issues will increase the scientific value of your publication. Best regards

Response: The conclusion is improved.